# Adaptive Correlated Monte Carlo for Contextual Categorical Sequence Generation

**Xinjie Fan**[1], **Yizhe Zhang**[2], **Zhendong Wang**[3], **Mingyuan Zhou**[1]

[1]University of Texas at Austin, [2]Microsoft Research, [3]Columbia University

`xfan@utexas.edu`, `yizhe.zhang@microsoft.com`,
`zw2533@columbia.edu`, `mingyuan.zhou@mccombs.utexas.edu`

## Abstract

Sequence generation models are commonly refined with reinforcement learning over user-defined metrics. However, high gradient variance hinders the practical use of this method. To stabilize this method, we adapt to contextual generation of categorical sequences a policy gradient estimator, which evaluates a set of correlated Monte Carlo (MC) rollouts for variance control. Due to the correlation, the number of unique rollouts is random and adaptive to model uncertainty; those rollouts naturally become baselines for each other, and hence are combined to effectively reduce gradient variance. We also demonstrate the use of correlated MC rollouts for binary-tree softmax models, which reduce the high generation cost in large vocabulary scenarios by decomposing each categorical action into a sequence of binary actions. We evaluate our methods on both neural program synthesis and image captioning. The proposed methods yield lower gradient variance and consistent improvement over related baselines.

## 1 Introduction

Contextual categorical sequence generation is a core modeling component in a wide variety of machine learning tasks, such as neural program synthesis (Bunel et al., 2018; Devlin et al., 2017b; Si et al., 2018; Chen et al., 2019) and image captioning (Vinyals et al., 2015; Xu et al., 2015). Typically, an encoder-decoder framework is applied. The encoder maps a contextual input to a latent representation, conditioning on which and previously generated tokens the decoder generates categorical tokens in a consecutive manner (Bahdanau et al., 2014; Sutskever et al., 2014; Cho et al., 2014; Rush et al., 2015; Chopra et al., 2016). It is common to train contextual sequence generation models using maximum likelihood estimation (MLE), which attempts to maximize the likelihood of each token in a target sequence given its preceding tokens. Learning with MLE is often sub-optimal as it does not directly optimize the evaluation metric of the end task. It generally suffers from the *exposure bias* (Bengio et al., 2015; Ranzato et al., 2016) , which refers to the discrepancy between training and generation using the Teacher Forcing (Williams & Zipser, 1989) strategy, *i.e.*, during training ground truth tokens are used as inputs, while during generation, only generated tokens are available. Thus giving higher likelihoods to target sequences does not guarantee the model to generate sequences close to the target or good sequences. Moreover, MLE requires target sequences for training, while for many scenarios in task-oriented dialogue (Williams & Young, 2007) and program synthesis (Zhong et al., 2017), only the final rewards to the generated sequences are available.

To overcome the aforementioned issues of MLE, it is common to refine a contextual sequence generation model pre-trained with MLE under the reinforcement learning (RL) framework (Zaremba & Sutskever, 2015; Ranzato et al., 2016; Bahdanau et al., 2016; Wu et al., 2018; Paulus et al., 2017). The objective becomes maximizing the expected rewards of model generated sequences. During training, only the model generated tokens are fed into the model so that the exposure bias is avoided. The reward to guide RL can be: 1) a task-dependent user-defined metric, such as CIDEr for image captioning (Vedantam et al., 2015) and *Generalization* for neural program synthesis (Bunel et al., 2018); and 2) automatically learned reward using a discriminator or language model (Yang et al., 2018; Yu et al., 2017; Lamb et al., 2016; Caccia et al., 2018; d'Autume et al., 2019). The RL training

---

Code link: https://github.com/xinjiefan/ACMC_ICLR

enables direct improvement of the user-defined or learned reward. Moreover, in cases where only weak-supervision is available, *e.g.*, in neural program synthesis, RL may considerably improve the model performance (Bunel et al., 2018; Zhong et al., 2017). However, the gradients of the expected reward in RL often suffer from high Monte Carlo (MC) estimation variance, due to noisy and/or sparse rewards and the large action space that grows exponentially with the sequence length.

There has been significant recent interest in variance reduction methods for MC gradient estimation (Mohamed et al., 2019). A highly effective solution is the reparameterization trick (Kingma & Welling, 2013; Rezende et al., 2014), which, however, is applicable to neither discrete variables nor non-differentiable reward functions. For variance reduction involving discrete variables, one potential solution is to combine the Gumbel-softmax trick, which relaxes the discrete variables to continuous ones, with reparameterization to produce low-variance but biased gradients (Jang et al., 2017; Maddison et al., 2017). Another common way for variance reduction is adding appropriate baselines (Owen, 2013; Williams, 1992; Paisley et al., 2012; Ranganath et al., 2014; Mnih & Gregor, 2014), and there exist several such methods customized for discrete variables (Tucker et al., 2017; Grathwohl et al., 2018). However, due to either the inherent biases or difficulty to learn the parameters of the baselines, it is unclear how effective these newly proposed estimators are in backpropagating the gradients through a sequence of discrete variables (Yin et al., 2019). This is exacerbated in contextual categorical sequence generation problems, where it is common for a sequence to contain quite a few tokens/actions, each of which is selected from a set of thousands of candidates.

Another practical issue is that generating a token from a large vocabulary via the softmax output layer is often computationally heavy. This prevents a categorical sequence generation model from being deployed to low-power devices. Despite significant recent efforts in addressing the computation bottleneck due to a wide softmax layer (Shim et al., 2017; Zhang et al., 2018; Chen et al., 2018), for categorical sequence generation, it is so far unclear how to address the softmax computational bottleneck while at the same time providing low-variance gradient of its RL objective.

This paper makes two primary contributions: 1) To address the high gradient variance issue, we adapt to contextual categorical sequence generation tasks the augment-REINFORCE-swap-merge (ARSM) estimator (Yin et al., 2019), which provides unbiased, low-variance gradients for categorical variables, using token-level rewards from correlated MC rollouts that naturally serve as the baselines for each other. We show that the number of rollouts is adapted to the model uncertainty on the latest generated token, and can also be manually controlled to balance variance reduction and computation. 2) To address the high generation cost issue, we replace the generation of each categorical variable in the sequence, via a categorical softmax output layer, with the generation of a sequence of binary decisions on a binary tree from its root node to a leaf node, which is occupied by a unique term of the vocabulary. For training, we adapt the augment-REINFORCE-merge (ARM) estimator (Yin & Zhou, 2019), which provides unbiased, low-variance gradients for binary variables, to backpropagate the gradients through the sequence of binary sequences. Under this binary-tree construction, the cost of generating a categorical token reduces from $O(V)$ to $O(\log_2(V))$, where $V$ is the vocabulary size.

We demonstrate our methods on two representative contextual categorical sequence generation tasks, with the number of actions ranging from 53 (neural program synthesis) to 9978 (image captioning).

## 2 PRELIMINARIES ON CONTEXTUAL SEQUENCE GENERATION

For a dataset of context-output pairs $\mathcal{D} := \{\boldsymbol{x}_i, \boldsymbol{y}_i\}_{i=1}^N$, our goal is to learn the conditional distribution of output $\boldsymbol{y}_i$ given its context $\boldsymbol{x}_i$, expressed as $p_{\boldsymbol{\theta}}(\boldsymbol{y}_i \,|\, \boldsymbol{x}_i)$. Below we drop the data index subscript for brevity. We focus on the case that an output is a sequence of $T$ categorical variable as $\boldsymbol{y} = \{y_1, \cdots, y_T\}$, where $y_t \in \{1, \ldots, V\}$. A common way to model $p_{\boldsymbol{\theta}}(\boldsymbol{y} \,|\, \boldsymbol{x})$ is to decompose it as $p_{\boldsymbol{\theta}}(\boldsymbol{y} \,|\, \boldsymbol{x}) = \prod_{t=1}^T p_{\boldsymbol{\theta}}(y_t \,|\, y_{1:t-1}, \boldsymbol{x})$, where the $t$-th term in the product, which models the distribution of token $y_t$ conditioning on the context $\boldsymbol{x}$ and previously generated tokens $y_{1:t-1}$, is commonly parameterized by a recurrent neural network (Sutskever et al., 2014). MLE is a common way to train the model: $\hat{\boldsymbol{\theta}}_{\text{MLE}} = \operatorname{argmax}_{\boldsymbol{\theta}} \mathbb{E}_{\{\boldsymbol{x}, \boldsymbol{y}\} \sim p_{\text{data}}(\boldsymbol{x}, \boldsymbol{y})}[\log p_{\boldsymbol{\theta}}(\boldsymbol{y} \,|\, \boldsymbol{x})]$. Viewing $p_{\boldsymbol{\theta}}(y_t \,|\, y_{1:t-1}, \boldsymbol{x})$ as a stochastic policy for choosing an action given the state, we can formulate contextual sequence generation as an RL problem and infer the policy parameter $\boldsymbol{\theta}$ as $\hat{\boldsymbol{\theta}}_{\text{RL}} = \operatorname{argmax}_{\boldsymbol{\theta}} \mathbb{E}_{\{\boldsymbol{x}, \boldsymbol{y}\} \sim p_{\text{data}}(\boldsymbol{x}, \boldsymbol{y})} \mathbb{E}_{\boldsymbol{z} \sim p_{\boldsymbol{\theta}}(\cdot \,|\, \boldsymbol{x})}[r(\boldsymbol{z} \,|\, \boldsymbol{x}, \boldsymbol{y})]$, where $r(\boldsymbol{z} \,|\, \boldsymbol{x}, \boldsymbol{y})$ denotes the reward of the generated (hypothesis) sequence $\boldsymbol{z}$ given the context $\boldsymbol{x}$ and the reference target sequence $\boldsymbol{y}$. For

example, for image captioning, the reward could be the CIDEr score that measures the similarity between the generated caption $z$ and the reference $y$ (Rennie et al., 2017).

Denote $\sigma(\cdot)$ as the softmax function and $\mathcal{T}_{\boldsymbol{\theta}}(\cdot)$ as a deterministic function defined by a deep neural network with parameter $\boldsymbol{\theta}$. We model $p_{\boldsymbol{\theta}}(\boldsymbol{z} \,|\, \boldsymbol{x}) = \prod_{t=1}^{T} p_{\boldsymbol{\theta}}(z_t \,|\, z_{1:t-1}, \boldsymbol{x})$, where

$$p_{\boldsymbol{\theta}}(z_t \,|\, \boldsymbol{x}, z_{1:t-1}) = \text{Cat}(\sigma(\boldsymbol{\phi}_t)), \quad \boldsymbol{\phi}_t := \mathcal{T}_{\boldsymbol{\theta}}(\boldsymbol{x}, z_{1:t-1}). \tag{1}$$

For a context-target pair $\{\boldsymbol{x}, \boldsymbol{y}\}$, we can expand the expected reward ER under policy $p_{\boldsymbol{\theta}}(\boldsymbol{z} \,|\, \boldsymbol{x})$ as

$$\text{ER} = \mathbb{E}_{\boldsymbol{z} \sim p_{\boldsymbol{\theta}}(\cdot \,|\, \boldsymbol{x})}[r(\boldsymbol{z} \,|\, \boldsymbol{x}, \boldsymbol{y})] = \mathbb{E}_{z_{1:t-1} \sim p_{\boldsymbol{\theta}}(\cdot \,|\, \boldsymbol{x})} \mathbb{E}_{z_t \sim \text{Cat}(\sigma(\boldsymbol{\phi}_t))}[r(z_{1:t} \,|\, \boldsymbol{x}, \boldsymbol{y})], \tag{2}$$

where the partial-sentence reward is defined as $r(z_{1:t} \,|\, \boldsymbol{x}, \boldsymbol{y}) = \mathbb{E}_{z_{t+1:T} \sim p_{\boldsymbol{\theta}}(\cdot \,|\, x, z_{1:t})}[r(\boldsymbol{z} \,|\, \boldsymbol{x}, \boldsymbol{y})]$.

Using the chain rule and REINFORCE (Williams, 1992) estimator, we have

$$\nabla_{\boldsymbol{\theta}} \text{ER} = \sum_{t=1}^{T} \nabla_{\boldsymbol{\phi}_t} \text{ER} \nabla_{\boldsymbol{\theta}} \boldsymbol{\phi}_t, \quad \nabla_{\boldsymbol{\phi}_t} \text{ER} = \mathbb{E}_{z_{1:t-1} \sim p_{\boldsymbol{\theta}}(\cdot \,|\, \boldsymbol{x})} \nabla_{\boldsymbol{\phi}_t} \mathbb{E}_{z_t \sim \text{Cat}(\sigma(\boldsymbol{\phi}_t))}[r(z_{1:t} \,|\, \boldsymbol{x}, \boldsymbol{y})],$$
$$\nabla_{\boldsymbol{\phi}_t} \mathbb{E}_{z_t \sim \text{Cat}(\sigma(\boldsymbol{\phi}_t))}[r(z_{1:t} \,|\, \boldsymbol{x}, \boldsymbol{y})] = \mathbb{E}_{z_t \sim \text{Cat}(\sigma(\boldsymbol{\phi}_t))}[r(z_{1:t} \,|\, \boldsymbol{x}, \boldsymbol{y}) \nabla_{\boldsymbol{\phi}_t} \ln p(z_t; \sigma(\boldsymbol{\phi}_t))].$$

The main challenge here is to control the variance in estimating $\nabla_{\boldsymbol{\theta}} \text{ER}$. A variety of methods have been proposed. For example, drawing sentence $\boldsymbol{z} \sim p_{\boldsymbol{\theta}}(\boldsymbol{z} \,|\, \boldsymbol{x})$ and using its reward $r(\boldsymbol{z} \,|\, \boldsymbol{x}, \boldsymbol{y})$ to approximate all partial-sentence rewards $\{r(z_{1:t} \,|\, \boldsymbol{x}, \boldsymbol{y})\}_{1,T}$, Ranzato et al. (2016) introduce MIXER with a scheduled training iterating between MLE and RL, estimating the RL gradient as

$$\hat{\nabla}_{\boldsymbol{\theta}} \text{ER} = \sum_{t=1}^{T} (r(\boldsymbol{z} \,|\, \boldsymbol{x}, \boldsymbol{y}) - b(z_{1:t-1})) \nabla_{\boldsymbol{\phi}_t} \ln p(z_t; \sigma(\boldsymbol{\phi}_t)) \nabla_{\boldsymbol{\theta}} \boldsymbol{\phi}_t,$$

where $b(z_{1:t-1})$ is a baseline function. To improve MIXER, Rennie et al. (2017) introduces the self-critic (SC) sequence training algorithm that sets $b(z_{1:t-1}) = r(\tilde{\boldsymbol{z}} \,|\, \boldsymbol{x}, \boldsymbol{y})$ for all $t$, where $\tilde{\boldsymbol{z}}$ is a greedy sequence rollout under $p_{\boldsymbol{\theta}}(\boldsymbol{z} \,|\, \boldsymbol{x})$. As using sentence-level reward $r(\boldsymbol{z} \,|\, \boldsymbol{x}, \boldsymbol{y})$ to guide the learning is found to be sensitive to the algorithm parameters such as the learning rate, Liu et al. (2017) follow Yu et al. (2017) to use token-level rewards that approximate each partial-sentence reward with $K$ independent MC rollouts (MC-$K$) as $\hat{r}(z_{1:t} \,|\, \boldsymbol{x}, \boldsymbol{y}) = \frac{1}{K} \sum_{k=1}^{K} r(z_{1:t}, z_{(t+1):T}^{(k)} \,|\, \boldsymbol{x}, \boldsymbol{y})$, $z_{(t+1):T}^{(k)} \sim p_{\boldsymbol{\theta}}(\cdot \,|\, \boldsymbol{x}, z_{1:t})$. With these token-level rewards $\hat{r}(z_{1:t} \,|\, \boldsymbol{x}, \boldsymbol{y})$, Liu et al. (2017) estimate the gradient as

$$\hat{\nabla}_{\boldsymbol{\theta}} \text{ER} = \sum_{t=1}^{T} (\hat{r}(z_{1:t} \,|\, \boldsymbol{x}, \boldsymbol{y}) - b(z_{1:t-1})) \nabla_{\boldsymbol{\phi}_t} \ln p(z_t; \sigma(\boldsymbol{\phi}_t)) \nabla_{\boldsymbol{\theta}} \boldsymbol{\phi}_t. \tag{3}$$

Following SC, for token $t$, one may choose baseline $b(z_{1:t-1}) = r(z_{1:t-1}, \tilde{z}_{t:T} \,|\, \boldsymbol{x}, \boldsymbol{y})$, where $\tilde{z}_{t:T}$ is a greedy rollout following partial sentence $z_{1:t-1}$. In addition to being more robust, using token-level rewards $\hat{r}(\boldsymbol{z}_{1:t})$ to guide the learning is often necessary when the reward signal is sparse, *e.g.*, in neural program synthesis, it is common that a full MC roll receives zero reward with high probability.

In addition to these methods mentioned above, the actor-critic method (Sutton, 1988) is used to reduce gradient variance at the expense of introducing bias (Bahdanau et al., 2016; Zhang et al., 2017). Several approaches explore beam search instead of sampling based methods (Wiseman & Rush, 2016; Bunel et al., 2018), also at the expense of introducing bias. Several other methods combine the MLE and RL objectives for training (Norouzi et al., 2016; Ding & Soricut, 2017).

## 3  POLICY GRADIENT WITH ADAPTIVE CORRELATED MC ROLLOUTS

Correlated MC samples, if well designed, can be combined to achieve much greater variance reduction than using the same number of independent ones (Owen, 2013). To reduce gradient variance for contextual categorical sequence generation and remove the need to construct explicit baselines, we adapt the augment-REINFORCE-swap (ARS) and ARS-merge (ARSM) estimators of Yin et al. (2019) to generate *correlated* MC rollouts. The number of correlated MC rollouts, used to estimate each token-level partial-sequence reward, is *adaptive* according to the uncertainty on token generation. The key idea here is to rewrite the gradient as differently expressed but equivalent expectations, whose MC samples, generated by sharing the same set of random numbers and hence correlated, are subsequently merged for variance reduction, without the need of learnable baseline functions.

Denote $z \sim \text{Cat}(\sigma(\boldsymbol{\phi}))$ as a univariate categorical variable such that $P(z = v \,|\, \boldsymbol{\phi}) = \sigma(\boldsymbol{\phi})_v = e^{\phi_v} / \sum_{i=1}^{V} e^{\phi_i}$. Denote $\boldsymbol{\pi} = (\pi_1, \ldots, \pi_V) \sim \text{Dir}(\mathbf{1}_V)$ as a random probability vector drawn from the Dirichlet distribution whose $V$ parameters are all ones. Denoting $m \leftrightarrows j$ as the operation of

swapping the $m$th and $j$th elements of a vector, we have $\pi_m^{m \leftrightharpoons j} = \pi_j$, $\pi_j^{m \leftrightharpoons j} = \pi_m$, and $\pi_i^{m \leftrightharpoons j} = \pi_i, \forall i \notin \{m, j\}$. With $z := \operatorname{argmin}_{i \in \{1, \cdots, V\}} \pi_i e^{-\phi_i}$, $\mathcal{E}(\boldsymbol{\phi}) = \mathbb{E}_{c \sim \operatorname{Cat}(\sigma(\boldsymbol{\phi}))}[r(c)]$ can be reexpressed as $\mathcal{E}(\boldsymbol{\phi}) = \mathbb{E}_{\boldsymbol{\pi} \sim \operatorname{Dir}(\mathbf{1}_V)}[r(z)]$, whose gradient under the ARS estimator can be expressed as

$$\nabla_{\phi_v} \mathcal{E}(\boldsymbol{\phi}) = \mathbb{E}_{\boldsymbol{\pi} \sim \operatorname{Dir}(\mathbf{1}_V)}[g_{\text{ARS}}(\boldsymbol{\pi}, j)_v], \quad g_{\text{ARS}}(\boldsymbol{\pi}, j)_v := [r(z^{v \leftrightharpoons j}) - \tfrac{1}{V} \sum_{m=1}^V r(z^{m \leftrightharpoons j})](1 - V\pi_j), \quad (4)$$

where $j$ is a reference category randomly selected from $\{1, \ldots, V\}$ and $z^{m \leftrightharpoons j} := \operatorname{argmin}_{i \in \{1, \cdots, V\}} \pi_i^{m \leftrightharpoons j} e^{-\phi_i}$. ARSM further improves ARS by adding a merge step as

$$\nabla_{\phi_v} \mathcal{E}(\boldsymbol{\phi}) = \mathbb{E}_{\boldsymbol{\pi} \sim \operatorname{Dir}(\mathbf{1}_V)}[g_{\text{ARSM}}(\boldsymbol{\pi})_v], \quad g_{\text{ARSM}}(\boldsymbol{\pi})_v := \tfrac{1}{V} \sum_{j=1}^V g_{\text{ARS}}(\boldsymbol{\pi}, j)_v. \quad (5)$$

We refer to $z$ as the true action and $z^{m \leftrightharpoons j}$ as pseudo actions. These actions are correlated to each other as they are transformed from the same Dirichlet distributed $\boldsymbol{\pi}$ vector under different pairwise index swaps. While there are $V(V-1)/2$ unique pairwise swaps, after MLE pre-train with $V \sim 10^4$, the number of unique pseudo actions that differ from the true action is random and often stays below 10, and becomes 0 more and more frequently as the progress of RL training reduces model uncertainty.

### 3.1 Adaptive correlated MC based policy gradient for categorical sequence

Applying the ARSM estimator in (5) to the expected reward shown in (2), we have

$$\nabla_{\phi_{tv}} \text{ER} = \mathbb{E}_{z_{1:t-1} \sim p_{\boldsymbol{\theta}}(\cdot \,|\, \boldsymbol{x})} \mathbb{E}_{\boldsymbol{\pi}_t \sim \operatorname{Dir}(\mathbf{1}_V)}[g_{\text{ARSM}}(\boldsymbol{\pi}_t)_v], \quad g_{\text{ARSM}}(\boldsymbol{\pi}_t)_v := \tfrac{1}{V} \sum_{j=1}^V g_{\text{ARS}}(\boldsymbol{\pi}_t, j)_v,$$

$$g_{\text{ARS}}(\boldsymbol{\pi}_t, j)_v := \big[ r(z_{1:t-1}, z_t^{v \leftrightharpoons j} \,|\, \boldsymbol{x}, \boldsymbol{y}) - \tfrac{1}{V} \sum_{m=1}^V r(z_{1:t-1}, z_t^{m \leftrightharpoons j} \,|\, \boldsymbol{x}, \boldsymbol{y}) \big] (1 - V\pi_{tj}),$$

where $z_t^{m \leftrightharpoons j} := \operatorname{argmin}_{i \in \{1, \cdots, V\}} \pi_{ti}^{m \leftrightharpoons j} e^{-\phi_{ti}}$, $\boldsymbol{\pi}_t \sim \operatorname{Dir}(\mathbf{1}_V)$, and $r(z_{1:t-1}, z_t^{m \leftrightharpoons j} \,|\, \boldsymbol{x}, \boldsymbol{y}) = \mathbb{E}_{z_{(t+1):T} \sim p_{\boldsymbol{\theta}}(\cdot \,|\, \boldsymbol{x}, z_{1:t-1}, z_t^{m \leftrightharpoons j})}[r(\boldsymbol{z} \,|\, \boldsymbol{x}, \boldsymbol{y})]$. We can therefore estimate each expectation using regular MC in the augmented space. Detailed formulations are deferred to Appendix B.1. Note if given $\boldsymbol{\pi}_t$, all pseudo actions $z_t^{m \leftrightharpoons j}$ are equal to true action $z_t$, then $g_{\text{ARSM}}(\boldsymbol{\pi}_t)_v = g_{\text{ARS}}(\boldsymbol{\pi}_t, j)_v = 0$.

We note while the notation appears cumbersome, its implementation is not difficult, as described in Algorithm 2, Appendix C. The intuitive explanation of ARSM is that given $\boldsymbol{\pi}_{1:T}$, it first generates true action sequence $z_{1:T}$ (main trajectory) with $z_t = \operatorname{argmin}_i \pi_{ti} e^{\phi_{ti}}$; it then performs embarrassingly parallel MC rollouts for all unique pseudo actions that differ from their corresponding true actions: at step $t$, given the true actions $z_{1:t-1}$, it generates pseudo actions $z_t^{m \leftrightharpoons j}$, and for each unique value of them that differs from $z_t$, it estimates its expected reward by rolling out a full sequence of length $T$; and finally it combines the sampled rewards of the true action sequence and unique pseudo action sequences, which are correlated to each other, to achieve significant variance reduction.

Despite significant gradient variance reduction, ARSM may become less efficient in computation when $V$ becomes large (*e.g.*, $\sim 10,000$). In the worst case, for each gradient estimate, it needs to generate as many as $V - 1$ unique pseudo action sequences at each token; while in practice, the actual number is much smaller, it still could be large enough to cause computational issues, especially if the policy parameter is far from convergence. We note while in theory ARSM enjoys embarrassingly parallel computation for rolling out all unique pseudo action sequences, the acceleration via parallelization in practice is constrained by the capacity of our own computation platform.

This motivates the following remedy. For large $V$, we choose $K$ reference categories $\gamma_1, \ldots, \gamma_K$, randomly sampled from $\{1, \ldots, V\}$ without replacement, to perform the swapping operations for pseudo action generation, and averaging over their corresponding ARS estimators as

$$g_{\text{ARS-K}}(\boldsymbol{\pi})_v = \tfrac{1}{K} \sum_{j=1}^K g_{\text{ARS}}(\boldsymbol{\pi}, \gamma_j)_v. \quad (6)$$

We refer to this gradient estimator as the ARS-K gradient estimator. Whether this remedy could be successful depends on how large $K$ needs to be as $V$ increases. We find via experiments that the sufficient size of $K$ grows slowly as $V$ increases. For example, we will show in Section 4.2 that for the image captioning task with $V = 9,788$, setting $K = 5$ already leads to competitive results.

Note during testing, regardless of whether using ARSM, ARS-K, or some other estimators, the categorical softmax output layer could become the computation bottleneck for random sequence generation. This motivates us to provide an algorithm to considerably reduce the generation cost during testing, though at the expense of reduced performance. We describe such a solution below.

## 3.2 BINARY-TREE-ARSM FOR COMPUTATIONAL RESOURCE LIMITED APPLICATIONS

The conventional way to generate a word token is to sample from a $V$-way categorical distribution, whose probability parameters are obtained via a softmax output layer. This softmax output layer often becomes the computation bottleneck when $V$ is large, making it difficult to be applied to resource-constrained environments, such as mobile devices. To mitigate this issue, related to the hierarchical softmax idea (Morin & Bengio, 2005; Grave et al., 2017; Goodman, 2001), we first construct a binary tree to allocate each word of the vocabulary to one and only one leaf node of this tree. A simple solution is to perform binary hierarchical clustering of the words.

Denote $\boldsymbol{e}_v$ as the word embedding vector of word $v$. In this paper, we use agglomerative clustering (Sibson, 1973) on $\boldsymbol{e}_1, \ldots, \boldsymbol{e}_V$ to recursively merge two closest clusters at a time until there is only one cluster. The root is linked to $V$ leaf nodes via $V$ overlapping root-to-leaf paths, each of which can be represented by a unique binary code $\boldsymbol{b}_v$ of length $D$, where $D = O(\log_2 V)$ is the depth of the tree. Note the $V$ paths are not restricted to travel through the same number of nodes, but for simplicity we zero pad them to the same length. Both off-the-shelf embedding vectors (Pennington et al., 2014) and task-specific ones can be utilized. They provide useful prior information about the structure of the vocabulary, which we can exploit to facilitate our search within the action space.

With the binary tree, we transform the problem of choosing one out of $V$ categories into that of making a sequence of binary decisions $\boldsymbol{b} = (b_1, \ldots, b_D)$. If making $l < D$ binary decisions $(b_1, \ldots, b_l)$ has already led to a leaf node, then the sequence is terminated and $b_{l+1}, \ldots, b_D$ all become zeros. There is a one-to-one mapping between the $V$ root-to-leaf paths and $V$ vocabulary words. We denote $\nu(\boldsymbol{b}) \in \{1, \ldots, V\}$ as the word that path $\boldsymbol{b}$ is mapped to, and $\boldsymbol{\beta}(v) \in \{0, 1\}^D$ as the path that word $v$ is mapped to. Note for a binary tree with $V$ leaves, there will be $V - 1$ non-leaf nodes, each of which needs a logit $\phi$ for its Bernoulli probability. Thus in total we need $V - 1$ logits $\phi_1, \cdots, \phi_{V-1}$. The computational saving in generating categorical sequences comes from the fact that to generate a word token we need $D$ $\phi$'s at most rather than all $V - 1$ $\phi$'s. Therefore, with the binary tree, the computation for the softmax output layer to generate a token decreases from $O(V)$ to $O(\log_2 V)$, which is significant especially for mobile applications. The binary-tree softmax model can be trained with MLE, or with the binary-tree-ARSM (BT-ARSM) gradient estimator introduced below.

For the binary case, both ARS and ARSM reduce to augment-REINFORCE-merge (ARM) (Yin & Zhou, 2019), which expresses the gradient of $\mathcal{E}_b(\phi) = \mathbb{E}_{z \sim \text{Ber}(\sigma(\phi))}[r(z)]$, $\sigma(\phi) = 1/(1 + e^{-\phi})$, as

$$\nabla_\phi \mathcal{E}_b(\phi) = \mathbb{E}_{\pi \sim \text{Uniform}(0,1)}[g_{\text{ARM}}(\pi)], \quad g_{\text{ARM}}(\pi) := [r(b_{\text{true}}) - r(b_{\text{sudo}})](1/2 - \pi), \qquad (7)$$

where $b_{\text{true}} := \mathbf{1}_{[\pi < \sigma(\phi)]}$ and $b_{\text{sudo}} := \mathbf{1}_{[\pi > \sigma(-\phi)]}$ are referred to as the true and pseudo actions, respectively. We note if we represent a $V$-way categorical variable as a sequence of $D = O(\log_2 V)$ binary variables, the number of unique pseudo actions that differ from the true actions is at most $D$.

In the binary-tree setting, the conditional probability of generating token $z_t$ is changed from (1) to

$$p_{\boldsymbol{\theta}}(z_t \,|\, \boldsymbol{x}, z_{1:t-1}) = \prod_{l=1}^{D_{z_t}} \text{Bernoulli}\left(b_{tl}; \sigma(\phi_{t, b_{t(1:l-1)}})\right), \quad (\phi_{t1}, \ldots, \phi_{t(V-1)}) := \mathcal{T}_{\boldsymbol{\theta}}(\boldsymbol{x}, z_{1:t-1}), \quad (8)$$

where $(b_{t1}, \ldots, b_{tD_{z_t}}) := \boldsymbol{\beta}(z_t)$, $\phi_{t, b_{t(1:l-1)}}$ is the parameter of the non-leaf node at the end of the path defined by $b_{t(1:l-1)}$, and $D_{z_t}$ is the number of non-leaf nodes in the root-to-leaf path that leads to $z_t$. Similar to the derivation in Section 3.1, we apply the ARSM gradient estimation to the decomposed binary sequences (BT-ARSM).

We provide the detailed formulations in Appendix B.2 and pseudo code in Algorithm 3, Appendix C. Intuitively, it first samples the true sequence of binary sequences $\{(b_{t1}, \ldots, b_{tD_{z_t}})\}_{t=1}^T$; it then performs embarrassingly parallel MC rollouts for all pseudo actions that differ from their corresponding true actions: at step $t$, and depth $l$, given the previous true tokens $z_{1:t-1}$, and true binary code $b_{t1}, \ldots, b_{t,l-1}$, it generates pseudo binary code $b_{tl}^{(\text{sudo})}$, and if it differs from $b_{tl}$, then we estimate its expected reward by first rolling out a full binary code up to depth $D$ and rolling out a full sequence up to length $T$; and finally it combines the sampled rewards of the true action sequence and pseudo action sequences, which are correlated to each other, to achieve significant variance reduction. Note that if $b_{tl}^{(\text{sudo})}$ is the same as $b_{tl}$, then the corresponding ARM gradient is zero.

## 4 EXPERIMENTS

We evaluate our models with both neural program synthesis (NPS) and image captioning.

Table 1: Comparison of various algorithms in terms of the *Generalization* score on the Karel dataset.

| Methods | MLE | SC | MC-0 | MC-2 | RL_beam | ARSM |
|---|---|---|---|---|---|---|
| *Generalization* (validation) | 13.6 | 12.51 | 12.64 | 13.56 | 14.76 | **17.07** |
| *Generalization* (test) | 12.76 | 12.12 | 12.56 | 12.76 | 14.92 | **16.28** |

## 4.1 NEURAL PROGRAM SYNTHESIS

NPS is a challenging representative task in contextual categorical sequence generation. First, the reward is only available after finishing the whole sequence. Second, the initial reward signals are often sparse because the generated programs rarely succeed in the beginning of training. We follow Bunel et al. (2018) to investigate an NPS task: for data sample $i$ consisting of a set of input-output states $\{I_i^m, O_i^m\}_{m=1,M_i}$, the goal is to learn a synthesizer parameterized by $\boldsymbol{\theta}$ to generate a program $\lambda_i$, which will produce a sequence of categorical actions to map input state $I_i^m$ to output state $O_i^m$ (*i.e.*, $\lambda_i(I_i^m) = O_i^m$) for all $m \in \{1, \ldots, M_i\}$. The evaluation metric is *Generalization* (Bunel et al., 2018), defined as the proportion of the test instances $\{I_{i'}^m, O_{i'}^m\}_{m=1,M_i}$ that satisfy $\lambda_{i'}(I_{i'}^m) = O_{i'}^m$ for all $m \in \{1, ..., M_i\}$. We evaluate on the Karel dataset (Devlin et al., 2017a), consisting of $10,000$ training reference Karel programs[1] with $2,500$ validation and $2,500$ test samples. Each program consists of a sequence of actions to move an agent inside a grid-world from one starting grid (input) to an end grid (output). The size of the action space $V$ is 53 and average program length is around 20.

**Baselines** We incorporate five baseline algorithms in our evaluation. ($i$) **MC-2** (Eq 3), using token-level rewards and greedy baselines. ($ii$) **MC-0**, using sentence-level reward and token-level greedy baselines, which corresponds to the TD-SCST in Rennie et al. (2017). ($iii$) **REINFORCE**, using sentence-level reward and with mini-batch mean as the baseline. ($iv$) **Self-Critic** (SC) as in Rennie et al. (2017). ($v$) **RL_beam**, the state-of-the-art method for NPS proposed by Bunel et al. (2018) to reduce the gradient variance while sacrificing the unbiasedness. The objective of RL_beam is to maximize the expected reward under a distribution defined on a space constructed with beam search $\mathrm{BS}(p_{\boldsymbol{\theta}}, S)$, where $S = 64$ is the beam size. Since the vocabulary size of $V = 53$ is not that large, we directly apply ARSM (*i.e.*, ARS-53) policy gradient and compare it with the other methods.

We use the code of Bunel et al. (2018) as basis and use the same model architecture except for the exclusion of the *optional* grammar checker. The grammar checker, not available for all NPS tasks, helps adaptively reduce the search (action) space and hence simplifies optimization. Excluding the optional grammar checker eliminates its confounding influence on the core NPS task, making the comparison more generic and fair. We use greedy search for both testing and validation. All policy gradient based methods are fine-tuning a pre-trained (and converged) MLE model.

**Results and analysis** Figs. 1a and 1b plot against iteration the log variance, and average number of rollouts (including greedy rollouts used to construct baselines) per step for each method. We observe that ARSM overall has the smallest gradient variance, and at the beginning ARSM has more MC rollouts (unique pseudo actions) and hence takes relative longer time per iteration, but soon it becomes more and more confident (reflected as fewer and fewer pseudo actions per iteration) and turns faster. We note that the gradient variance at a given iteration is related to both the property of the gradient estimator and the parameter value at that iteration. Thus having smaller gradient variance may not necessarily imply better performance if different learning algorithms are not moving their parameters towards the same solution. This could help explain why SC has lower gradient variance than both MC-0 and MC-2 do but worse validation and test *Generalization* scores.

Figs. 1c and 1d plot the *Generalization* scores against training time on the training and validation sets. Due to large gradient variance, all methods except ARSM and RL_beam either diverge or fail to improve the training objective. Examining the performance on the training and validation sets suggests that REINFORCE and SC both diverge quickly; MC-0 stays around the starting point; MC-2 improves upon MLE initially, but then gradually diverges; RL_beam reaches a good solution very fast but then gradually degrades towards worse solutions; and ARSM is the only one that makes steady improvement as the training progresses. Observing how the gap between training and testing evolves, we see evidence suggesting that RL_beam overfits the training data, possibly due to the use of biased gradients, while ARSM does not. We note that there is no explicit regularization in

---

[1]The original dataset contains 1 million training instances. Bunel et al. (2018) proposed to reduce the dataset to $10,000$ examples and observed significant improvement of RL upon MLE when the reference program data is limited. Our experiments are based on the same reduced dataset.

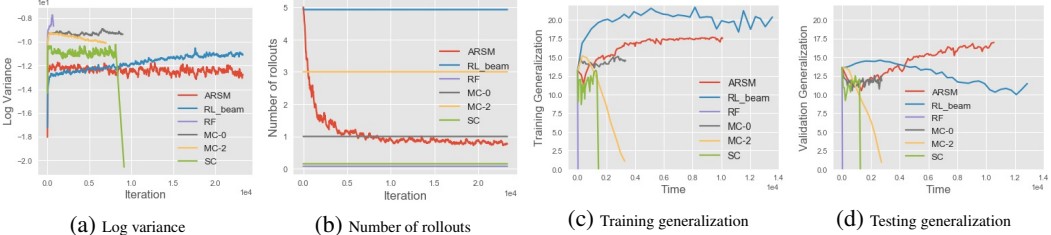

|(a) Log variance | (b) Number of rollouts | (c) Training generalization | (d) Testing generalization |

Figure 1: From left to right are the comparisons of various methods in terms of gradient variance, number of sequence rollouts, training *Generalization* score, and validation *Generalization* score.

ARSM. However, since ARSM tends to generate fewer and fewer unique pseudo actions as the policy becomes more and more confident, this adaptive characteristic may serve as an implicit regularization during the training process. Moreover, as the policy becomes more confident, the ARSM estimator has an increasing probability to yield MC gradient estimates that are exactly zeros, which may also help prevent overfitting as zero gradients will freeze the update of model parameters. We summarize the validation and test *Generation* scores in Table 1. Both MLE and RL_beam (Bunel et al., 2018) perform reasonably well, but are outperformed by ARSM with a large margin. Even though MC-2 seems to improve upon MC-0 and SC, indicating the importance of using token-level rewards rather than sentence-level reward to guide the learning in this sparse reward scenario, it still clearly underperforms ARSM, which on average uses much fewer rollouts to estimate token-level rewards. This demonstrates the advantage of using an adaptive number of correlated MC rollouts over a fixed number of independent MC rollouts.

## 4.2 IMAGE CAPTIONING

Image captioning, mapping an image $x$ to a summary sentence $y = (y_1, \ldots, y_T)$, has become a standard task to compare different policy gradient based RL methods. We conduct our experiments on the MS COCO dataset (Lin et al., 2014), following the standard data split from Karpathy & Fei-Fei (2015). We fine-tune a pre-trained MLE model using CIDEr score as the reward. Our implementation is based on Luo et al. (2018). Details about the experimental setup can be found in Appendix D.

**ARS-K for computation-sufficient deployments** We first investigate the effectiveness of the proposed method when it is computationally feasible to use the categorical softmax output layer at the test time. We consider MLE, REINFORCE, SC, and ARS-K with the same vocabulary size of $V = 9788$. For ARS-K, we experiment with several different $K$ values. We report CIDEr score, and other commonly used metrics for the test set in Table 2. We observe that while

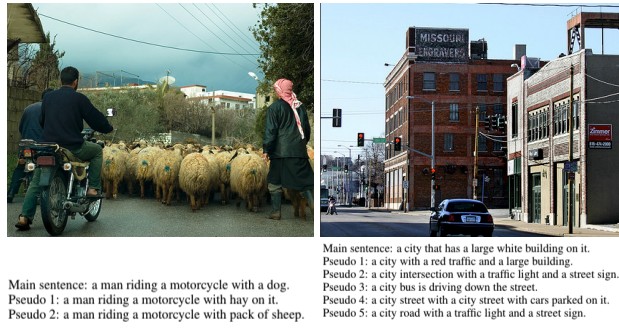

Figure 2: Main and pseudo trajectories for image captioning.

ARS-1 underperforms SC, ARS-K quickly improves as $K$ increases: ARS-5 becomes comparable to SC in performance; ARS-10 and ARS-20 outperform SC by a large margin with statistical significance (standard error is about 0.2). The superior performance of ARS-K with large $K$ is also evidenced by Fig. 3. The gradient variance of ARS-20 is significantly lower than other algorithms (Fig. 3b upper). In Fig. 3c (upper), we compare the average number of correlated MC rollouts of ARS-K for different $K$. While in theory the number of *unique* pseudo actions in ARS-K could be as many as $V - 1$ at each step, it can be seen that after MLE pre-training, for each ARS-K ($K = 1, 5, 10, 20$), on average that number is small (fewer than 10 for $V = 9488$ and $K = 20$) and has an evident decreasing trend during training. Moreover, it increases slowly as $K$ increases (clearly below a linear increasing rate).

Fig. 2 shows two pictures (see more plots in Appendix E.1) with their main sentences, which are greedily generated, and pseudo sentences generated by ARS-5 starting from the 7th token and 3rd token, respectively. Both greedily generated captions contain incorrect information about given

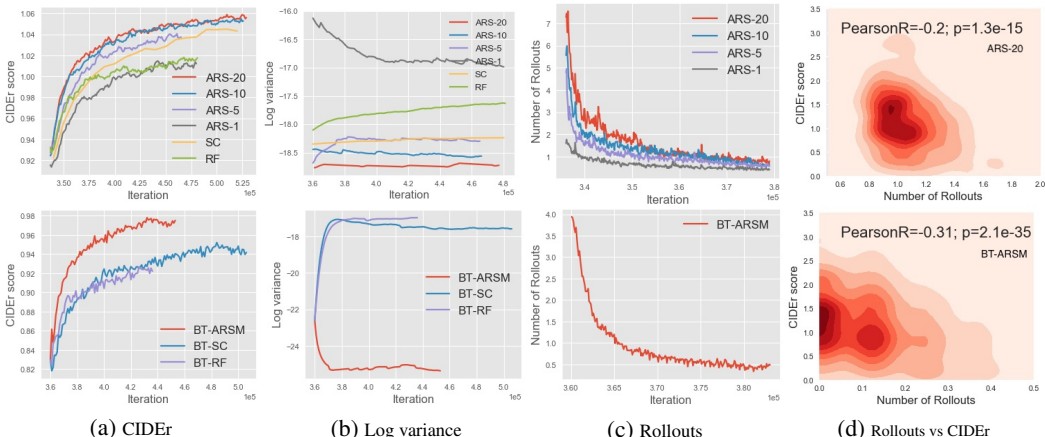

(a) CIDEr      (b) Log variance      (c) Rollouts      (d) Rollouts vs CIDEr

Figure 3: Comparison of different gradient estimators for image captioning task. "RF" denotes REINFORCE and "SC" denotes Self-Critic. Upper (lower) row: models using a regular softmax (binary-tree softmax).

images, while the pseudo sentences are semantically close to the greedy generations, however with interpretable variations in some details. Some pseudo sentences are better than the greedily generated captions. These pseudo-captions assembled together capture the nuance variations of the neighborhood of the generation, thus can serve as a good baseline to reduce the variance of the policy gradient. Note that there are more pseudo actions in the second plot, because the image is more complex and also there is more uncertainty at the beginning stage of generation (3rd token) compared to the latter stage of generation (7th token).

**BT-ARSM for computation-limited deployments** We evaluate the binary-tree ARSM (BT-ARSM) described in Section 3.2, which decomposes the action space to a sequence of binary actions.

**(a)** *Binary tree constructions and MLE pretraining*. We explore three different ways to construct binary trees over the action space: $(i)$ *tree_WV* : We apply agglomerative clustering to off-the-shelf pre-trained Word-to-Vector (WV) embeddings (Mikolov et al., 2013) to get a binary tree with a depth of 25; $(ii)$ *tree_DIS*: We follow tree_WV except that we use the word embeddings pre-trained for image captioning with standard MLE objective and full vocabulary to distill (DIS) the full-softmax model's knowledge to the binary tree; $(iii)$ *tree_RD* : We randomly permute the leaves of *tree_DIS* to produce a tree with no meaningful structure, referred as tree_RD. We pre-train all these three models with MLE, and report the CIDEr score in Table 2. Among these three binary trees, tree_DIS performs the best, indicating that the tree structure has impact on its performance, and a task-specific pre-trained embedding is preferable when constructing a binary tree. As expected, comparing with the models trained using regular softmax layer (Table 2), the performance of the models with binary-tree softmax layers drop. However, the Multi-Adds softmax operations needed for generating a token is reduced by $V/D$ ($\sim 380$ in our case) times, leading to a significant improvement in efficiency especially for deployment in *computing resource limited scenarios* at the cost of moderately degraded accuracy.

**(b)** *Fine-tuning using BT-ARSM*. We further fine-tune the pre-trained tree_DIS model, with binary-tree-REINFORCE (BT-RF), binary-tree-Self-Critic (BT-SC), and binary-tree-ARSM (BT-ARSM) respectively. Table 2 shows that BT-ARSM significantly outperforms the other two, which can be explained by the considerable variance reduction of BT-ARSM as is shown in Fig 3b (lower). Notably, the performance of BT-ARSM is superior to vanilla softmax model trained with MLE even though it has been injected with strong inductive bias via binary-tree softmax to reduce its generation cost.

**Adaptiveness of ARS-K and BT-ARSM** As shown in Fig. 3 and Fig. 4 (in Appendix A), our proposed methods can adaptively choose the number of correlated MC rollouts in four aspects: $(i)$(adapt across samples) in Fig. 3d, we show the 2-D density estimation for the numbers of rollouts and the CIDEr scores of different samples during the later stage of training. We observe a statistically significant negative correlation ($p < 0.05$) between the numbers of rollouts and CIDEr scores, indicating that our algorithms can adaptively generate more rollouts for harder samples (lower CIDEr scores) and less rollouts for easier ones (higher CIDEr scores); $(ii)$(adapt across iterations) as shown in Fig. 3c, during the training, the number of correlated MC rollouts decreases as the model improves and converges; $(iii)$(adapt across sentence positions) as shown in Figs. 4a, 4b, 4d, and 4e, more MC

Table 2: Performance comparison on the test set of COCO-caption dataset.

| Method | CIDEr | BLEU-4 | BLUE-3 | BLEU-2 | BLEU-1 | ROUGE | METEOR |
|---|---|---|---|---|---|---|---|
| Soft Attention (Xu et al., 2015) | – | 24.3 | 34.4 | 49.2 | 70.7 | – | 23.9 |
| Hard Attention (Xu et al., 2015) | – | 25.0 | 35.7 | 50.4 | 71.8 | – | 23.0 |
| Show & Tell (Vinyals et al., 2015) | 85.5 | 27.7 | – | – | – | – | 23.7 |
| ATT-FCN (You et al., 2016) | – | 30.4 | 40.2 | 53.7 | 70.9 | – | 24.3 |
| SCN-LSTM (Gan et al., 2017) | 101.2 | **33.0** | 43.3 | 56.6 | 72.8 | – | 25.7 |
| Vanilla Softmax with MLE | 93.3 | 30.4 | 40.2 | 53.6 | 70.7 | 52.2 | 24.7 |
| REINFORCE | 103.6 | 31.6 | 42.9 | 57.8 | 74.8 | 54.0 | 25.1 |
| Self-Critic (Rennie et al., 2017) | 106.5 | 32.3 | **43.8** | 58.7 | 75.6 | 54.6 | 25.6 |
| ARS-1 | 103.6 | 30.8 | 42.4 | 57.8 | 75.4 | 54.0 | 25.2 |
| ARS-5 | 106.1 | 31.6 | 43.3 | 58.6 | 76.1 | 54.6 | 25.5 |
| ARS-10 | 107.7 | 31.8 | 43.5 | 58.8 | 76.0 | 54.7 | 25.7 |
| ARS-20 | **108.4** | 32.1 | **43.8** | **59.0** | **76.4** | **54.8** | **25.8** |
| tree_RD with MLE | 77.5 | 23.1 | 33.7 | 48.6 | 67.1 | 49.2 | 22.4 |
| tree_WV with MLE | 80.2 | 23.3 | 34.1 | 49.3 | 68.0 | 49.9 | 22.9 |
| tree_DIS with MLE | 84.2 | 24.9 | 35.1 | 49.7 | 67.6 | 50.5 | 23.7 |
| tree_DIS with BT-RF | 93.6 | 28.8 | 39.7 | 55.0 | 72.5 | 52.2 | 23.7 |
| tree_DIS with BT-SC | 96.5 | 29.4 | 40.5 | 55.4 | 72.7 | **52.8** | 24.1 |
| tree_DIS with BT-ARSM | **99.2** | **29.9** | **41.2** | **56.2** | **73.3** | 52.7 | **24.3** |

rollouts appear at the timesteps in the middle range of the generated sequence, as they are associated with higher uncertainty; $(iv)$(adapt across depths) as shown in Figs. 4c and 4f, for binary-tree softmax, the top layers (close to the root) of the tree are associated with more MC rollouts, since they are more uncertain about what to predict. More details are provided in Appendix A.

# 5 CONCLUSION

In this paper, we demonstrate the adaptation of ARSM policy gradient estimator, utilizing token-level rewards of correlated Monte Carlo (MC) rollouts, to optimize contextual categorical sequence generation model. We apply the gradient estimators based on this idea to both the regular softmax model and binary-tree softmax model. The binary-tree softmax model has low cost for generating categorical tokens and hence is suited for computation-limited scenarios. We conduct empirical study on two challenging tasks: neural program synthesis and image captioning. Our observations verify that fewer and fewer correlated MC rollouts are conducted as the model becomes increasingly more certain during training. In addition, we show with correlated MC rollouts serving as baselines for each other, our methods show significant reduction of gradient variance and consistently outperform related baselines. We note that in a cold-start setting where we start from a complete random policy, it is still challenging to make our methods work efficiently as the number of pseudo actions may be too large if $V$ is large. We consider it as future work to adapt our methods to this more challenging setting, where, to our best knowledge, little work has been done except for Ding & Soricut (2017) and d'Autume et al. (2019).

ACKNOWLEDGMENTS

This research was supported in part by the U.S. National Science Foundation under Grant IIS-1812699 and the McCombs Research Excellence Grant. The authors acknowledge the support of NVIDIA Corporation with the donation of the Titan Xp GPU used for this research, and the computational support of Texas Advanced Computing Center.

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

# A    ADAPTIVENESS OF ARS-K/BT-ARSM

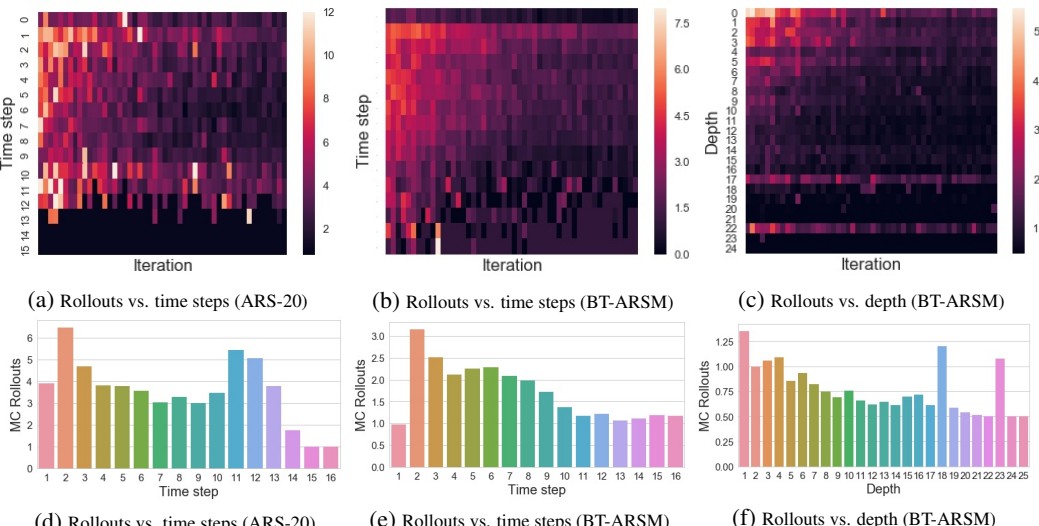

(a) Rollouts vs. time steps (ARS-20)  (b) Rollouts vs. time steps (BT-ARSM)  (c) Rollouts vs. depth (BT-ARSM)

(d) Rollouts vs. time steps (ARS-20)  (e) Rollouts vs. time steps (BT-ARSM)  (f) Rollouts vs. depth (BT-ARSM)

Figure 4: Adaptiveness of MC rollouts with BT-ARSM/ARS-K.

1. Adaptiveness across samples: In Fig. 3d, we show the 2-D density estimation for the numbers of rollouts and the CIDEr scores for different samples during the late stage of training. We observe a statistically significant negative correlation ($p < 0.05$) between the number of rollouts and CIDEr scores, indicating that our algorithms can adaptively generate more rollouts for harder samples (lower CIDEr scores) and less rollouts for easier samples (higher CIDEr scores).

2. Adaptiveness across time steps: As shown in Figs. 4a, 4b, 4d, and 4e, it is reasonable that there will be more MC rollouts at the middle time step of the generated sequence, since the initial words and end words are more easily to be learned due to their cardinality, and ARSM model will be more confident about the choices for these words, leading to fewer MC rollouts.

3. Adaptiveness across depths: For binary softmax model, the terms in the vocabulary are represented as leaf nodes in the tree. As shown in Figs. 4c and 4f, ASRM successfully captures the adaptiveness of MC rollouts across depths in the tree. The top layers (close to the root) of the tree are associated with more MC rollouts, since they are more uncertain about what to predict.

4. Adaptiveness across iterations: In Figs. 1b and 3c, we observe that, during training, the number of correlated MC rollouts decreases as the model improves and converges.

# B    DETAILED FORMULATIONS

## B.1    ARS/M GRADIENT ESTIMATORS

Applying the ARSM estimator in (5) to the expected reward shown in (2), we have

$$\nabla_{\phi_{tv}} \text{ER} = \mathbb{E}_{z_{1:t-1} \sim p_{\boldsymbol{\theta}}(\cdot \,|\, \boldsymbol{x})} \mathbb{E}_{\boldsymbol{\pi}_t \sim \text{Dir}(\mathbf{1}_V)} [g_{\text{ARSM}}(\boldsymbol{\pi}_t)_v], \quad g_{\text{ARSM}}(\boldsymbol{\pi}_t)_v := \frac{1}{V} \sum_{j=1}^{V} g_{\text{ARS}}(\boldsymbol{\pi}_t, j)_v,$$

$$g_{\text{ARS}}(\boldsymbol{\pi}_t, j)_v := \left[ r(z_{1:t-1}, z_t^{v \rightleftharpoons j} \,|\, \boldsymbol{x}, \boldsymbol{y}) - \frac{1}{V} \sum_{m=1}^{V} r(z_{1:t-1}, z_t^{m \rightleftharpoons j} \,|\, \boldsymbol{x}, \boldsymbol{y}) \right] (1 - V \pi_{tj}),$$

where $z_t^{m \rightleftharpoons j} := \text{argmin}_{i \in \{1, \cdots, V\}} \pi_{ti}^{m \rightleftharpoons j} e^{-\phi_{ti}}$. Thus we can approximate the gradient $\nabla_{\boldsymbol{\theta}} \text{ER}$ as

$$\hat{\nabla}_{\boldsymbol{\theta}} \text{ER} = \sum_{t=1}^{T} \sum_{v=1}^{V} \hat{g}_{\text{ARSM}}(\boldsymbol{\pi}_t)_v \nabla_{\boldsymbol{\theta}} \phi_{tv}, \quad \hat{g}_{\text{ARSM}}(\boldsymbol{\pi}_t)_v := \frac{1}{V} \sum_{j=1}^{V} \hat{g}_{\text{ARS}}(\boldsymbol{\pi}_t, j)_v,$$

$$\hat{g}_{\text{ARS}}(\boldsymbol{\pi}_t, j)_v := \left[ \hat{r}(z_{1:t-1}, z_t^{v \rightleftharpoons j} \,|\, \boldsymbol{x}, \boldsymbol{y}) - \frac{1}{V} \sum_{m=1}^{V} \hat{r}(z_{1:t-1}, z_t^{m \rightleftharpoons j} \,|\, \boldsymbol{x}, \boldsymbol{y}) \right] (1 - V \pi_{tj}), \quad (9)$$

where $\boldsymbol{\pi}_1, \ldots, \boldsymbol{\pi}_T \overset{iid}{\sim} \text{Dir}(\mathbf{1}_V)$; $\hat{r}(z_{1:t-1}, z_t^{m \rightleftharpoons j} \,|\, \boldsymbol{x}, \boldsymbol{y})$ is an approximation of $r(z_{1:t-1}, z_t^{m \rightleftharpoons j} \,|\, \boldsymbol{x}, \boldsymbol{y}) = \mathbb{E}_{z_{(t+1):T} \sim p_{\boldsymbol{\theta}}(\cdot \,|\, \boldsymbol{x}, z_{1:t-1}, z_t^{m \rightleftharpoons j})} [r(\boldsymbol{z} \,|\, \boldsymbol{x}, \boldsymbol{y})]$, which can be estimated with $r(z_{1:t-1}, z_t^{m \rightleftharpoons j}, \tilde{z}_{t+1:T} \,|\, \boldsymbol{x}, \boldsymbol{y})$, where $\tilde{z}_{t+1:T} \sim p_{\boldsymbol{\theta}}(\cdot \,|\, \boldsymbol{x}, z_{1:t-1}, z_t^{m \rightleftharpoons j})$ is an MC rollout.

## B.2    BT-ARSM GRADIENT ESTIMATORS

$$\nabla_{\phi_{t,b_{t(1:l-1)}}}\text{ER} = \mathbb{E}_{z_{1:t-1}\sim p_{\boldsymbol{\theta}}(\cdot\,|\,\boldsymbol{x})}\mathbb{E}_{b_{t(1:l-1)}\sim p_{\boldsymbol{\theta}}(\cdot\,|\,\boldsymbol{x},z_{1:t-1})}\mathbb{E}_{\pi_{tl}\sim\text{Unif}(0,1)}[g_{\text{ARM}}(\pi_{tl})],$$

$$g_{\text{ARM}}(\pi_{tl}) = \left[r\big(z_{1:t-1},b_{t(1:l-1)},b_{tl}^{(\text{true})}\,|\,\boldsymbol{x},\boldsymbol{y}\big) - r\big(z_{1:t-1},b_{t(1:l-1)},b_{tl}^{(\text{sudo})}\,|\,\boldsymbol{x},\boldsymbol{y}\big)\right](1/2 - \pi_{tl}),$$

$$b_{tl}^{(\text{true})} := \mathbf{1}_{[\pi_{tl}<\sigma(\phi_{t,b_{t(1:l-1)}})]},\ b_{tl}^{(\text{sudo})} := \mathbf{1}_{[\pi_{tl}>\sigma(-\phi_{t,b_{t(1:l-1)}})]}, \tag{10}$$

where $r(z_{1:t-1},b_{t(1:l)}\,|\,\boldsymbol{x},\boldsymbol{y}) := \mathbb{E}_{b_{t(l+1:D_{z_t})},z_{(t+1):T}\sim p_{\boldsymbol{\theta}}(\cdot\,|\,\boldsymbol{x},z_{1:t-1},b_{t(1:l)})}[r(\boldsymbol{z}\,|\,\boldsymbol{x},\boldsymbol{y})]$. Thus we have $\nabla_{\boldsymbol{\theta}}\text{ER} = \sum_{t=1}^{T}\sum_{l=1}^{D_{z_t}}\nabla_{\phi_{t,b_{t(1:l-1)}}}\text{ER}\,\nabla_{\boldsymbol{\theta}}\phi_{t,b_{t(1:l-1)}}$, which can be approximated with $\hat{\nabla}_{\boldsymbol{\theta}}\text{ER} = \sum_{t=1}^{T}\sum_{l=1}^{D_{z_t}}\hat{g}_{\text{ARM}}(\pi_{tl})(1-2\pi_{tl})$, where $\hat{g}_{\text{ARM}}(\pi_{tl})$ approximates $g_{\text{ARM}}(\pi_{tl})$ shown in (10) via MC integration; note if given $\pi_{tl}\sim\text{Unif}(0,1)$, $b_{tl}^{(\text{sudo})} = b_{tl}^{(\text{true})}$, then $g_{\text{ARM}}(\pi_{tl}) = 0$.

## C    ALGORITHMS

**Efficient Pseudo-Action Computation:** To implement ARS-K, we need to compute the corresponding pseudo actions for each reference category $j$ in a reference set $J$. In other words, given $\boldsymbol{\pi},\boldsymbol{\phi}$, we need to compute $z^{m\rightleftharpoons j} = \text{argmin}_{\{i=1,\ldots,V\}}\ln\pi_i^{m\rightleftharpoons j} - \phi_i$, for all $m\in\{1,\ldots,V\},j\in J$. A naive implementation would involve taking the minimum over $V$-dimensional vectors for $V\cdot|J|$ times, which is computationally expensive when $V$ is large. In the following, we take advantage of the correlation among pseudo actions and propose an efficient algorithm which only involves taking the minimum over $V$-dimensional vectors $|J|$ times. We first explain the notations and the basic ideas, and then present the algorithm in Algorithm 1.

Let $o_{i,j} = \ln\pi_i - \phi_j$. Denote $m_1,m_2$ as the indexes for the top 2 smallest in $\{o_{i,i}\}_{i=1:V}$ respectively (this is the only place where we need to take the minimum over whole $V$-dimensional vectors). Let $\text{ID}(o_{i,j})$ denote the function returning the second index of $o_{i,j}$, which means

$$\text{ID}(o_{i,j}) = j.$$

In the following, we show that it is not necessary to take minimum over whole $V$-dimensional vectors for each pseudo actions. We can compute $z^{m\rightleftharpoons j}$ efficiently by observing:

1. If $m_1\notin\{m,j\}$, the smallest value in $\ln\boldsymbol{\pi}^{m\rightleftharpoons j} - \boldsymbol{\phi}$ can only be among the updated two values $(o_{j,m},o_{m,j})$ and the original smallest value $o_{m_1,m_1}$. Hence, the index of the smallest value is

$$z^{m\rightleftharpoons j} = \text{ID}(\min(\min(o_{j,m},o_{m,j}),o_{m_1,m_1})).$$

2. If $m_1\in\{m,j\}$, $o_{m_1,m_1}$ will be updated to a new value, therefore, the above equation does not hold. But, in the following, we show we can use the second smallest value instead. The index of the smallest value will be

$$z^{m\rightleftharpoons j} = \text{ID}(\min(\min(o_{j,m},o_{m,j}),o_{m_2,m_2}))$$

*Proof.* Assume $m_1\in\{m,j\}$. If $m_2\notin\{m,j\}$, since $o_{m_1,m_1}$ has been changed, $o_{m_2,m_2}$ will be the smallest value in the vector excluding the two updated values. Hence, the smallest value will be among the three values $\{o_{j,m},o_{m,j},o_{m_2,m_2}\}$, among which we can find the index of the smallest item. If $m_2\in\{m,j\}$, then $\{m,j\}$ becomes $\{m_1,m_2\}$. Hence, $o_{m_1,m_1}$ and $o_{m_2,m_2}$ will be updated. For any $m\notin\{m_1,m_2\}$, $\min(o_{m_2,m_1},o_{m_1,m_2})\le o_{m_2,m_2}\le o_{m,m}$. Therefore, the smallest value will be among $\{o_{m_2,m_1},o_{m_1,m_2}\}$. The index of the smallest value will be $\text{ID}(\min(o_{m_2,m_1},o_{m_1,m_2}))$, which is equivalent to $\text{ID}(\min(\min(o_{j,m},o_{m,j}),o_{m_2,m_2}))$. $\square$

**Algorithm 1:** Compute Pseudo-Action Matrix for Reference Category Set J in Parallel

**input** : Batched $\boldsymbol{\pi}$ and $\boldsymbol{\phi}$, Ref-Cat Set $J$, ID Function
**output :** Pseudo-Action Matrix $P$;

Compute $o_{m_1}, m_1, o_{m_2}, m_2 = \text{Top2}(\ln \boldsymbol{\pi} - \boldsymbol{\phi})$;
**for** $j \in J, m \in \{1, \dots, V\}$ *(in parallel)* **do**
 Compute $o_{j,m} = \ln \pi_j - \phi_m$
 Compute $o_{m,j} = \ln \pi_m - \phi_j$
**end for**
Initialize $P$ with size $(|J|, V)$
**for** $j \in J, m \in \{1, \dots, V\}$ *(in parallel by using index matrix)* **do**
 **if** $m_1 \in \{j, m\}$ **then**
  $P[j, m] = \text{ID}(\min(\min(o_{j,m}, o_{m,j}), o_{m_1,m_1}))$
 **else**
  $P[j, m] = \text{ID}(\min(\min(o_{j,m}, o_{m,j}), o_{m_2,m_2}))$
 **end if**
**end for**

**Algorithm 2:** ARS-$K$/ARSM($K = V$) policy gradient for fine-tuning a contextual categorical sequence generation model with a discrete-action space of $V$ actions.

**input** : MLE pre-trained policy parameter $\boldsymbol{\theta}$, number of reference category $K$, main trajectory sample type $mt$, pseudo trajectory sample type $pt$

**output** : Fine-tuned policy parameter $\boldsymbol{\theta}$

**while** *not converged* **do**

  Receive random sample $\boldsymbol{x}, \boldsymbol{y}$;

  First, we sample a main trajectory $(z_1, \ldots, z_T)$:

  **if** $mt =$ *'greedily sample'* **then**

    for $t = 1 : T$, let $z_t = \operatorname{argmin}_{i \in \{1,\ldots,V\}}(-\phi_{ti})$, where $\boldsymbol{\phi}_t = \mathcal{T}_{\boldsymbol{\theta}}(\boldsymbol{z}_{1:t-1}, \boldsymbol{x})$;

  **else**

    for $t = 1 : T$, let $z_t = \operatorname{argmin}_{i \in \{1,\ldots,V\}}(\ln \pi_{ti} - \phi_{ti})$, where $\boldsymbol{\pi}_t \sim \text{Dirichlet}(\mathbf{1}_V)$ (or let $\pi_{ti} = -\ln(Unif(0,1)))$, and $\boldsymbol{\phi}_t = \mathcal{T}_{\boldsymbol{\theta}}(\boldsymbol{z}_{1:t-1}, \boldsymbol{x})$;

  **end if**

  Second, we compute pseudo actions:

  **for** $t = 1 : T$ **do**

    Let $\boldsymbol{\pi}_t \sim \text{Dirichlet}(\mathbf{1}_V)$ (or let $\pi_{ti} = -\ln(Unif(0,1))$ for $i = 1, \ldots, V$ and then normalize them to have a unit norm);

    Let $j_1, ..., j_K$ be $K$ reference categories randomly sampled from $\{1, \ldots, V\}$ without replacement;

    **for** $k = 1, \ldots, K$, $v = 1, \ldots, V$ *(in parallel)* **do**

      Let $z_t^{v \leftrightharpoons j_k} := \operatorname{argmin}_{i \in \{1,\ldots,V\}}(\ln \pi_{ti}^{v \leftrightharpoons j_k} - \phi_{ti})$ as the $(v, k)$th pseudo action;

    **end for**

    Let $S_t = \text{unique}(\{z_t^{v \leftrightharpoons j}\}_{v,j})$ which means $S_t$ is the set of all unique values in $\{z_t^{v \leftrightharpoons j}\}_{v,j}$. Denote the cardinality of $S_t$ as $|S_t|$, where $1 \leq |S_t| \leq V$ ;

  **end for**

  Third, we complete sentences and evaluate the reward for the unique set of pseudo actions:

  **for** $t = 1 : T$ *(in parallel)* **do**

    **if** $|S_t| = 1$ **then**

      continue

    **end if**

    **for** $\tilde{z}_{ts} \in S_t$ *(in parallel)* **do**

      **if** $t < T$ **then**

        **if** $pt =$ *'greedily sample'* **then**

          greedily sample $\boldsymbol{z}_{t+1:T}^s \sim p_{\boldsymbol{\theta}}(\boldsymbol{z}_{t+1:T} \mid \boldsymbol{z}_{1:t-1}, \tilde{z}_{ts}, \boldsymbol{x})$

        **else**

          randomly sample $\boldsymbol{z}_{t+1:T}^s \sim p_{\boldsymbol{\theta}}(\boldsymbol{z}_{t+1:T} \mid \boldsymbol{z}_{1:t-1}, \tilde{z}_{ts}, \boldsymbol{x})$

        **end if**

      **end if**

    **end for**

    **for** $v = 1 : V, k = 1 : K$ *(in parallel)* **do**

      Let $f(z_t^{v \leftrightharpoons j_k}) = r(\boldsymbol{z}_{1:t-1}, \tilde{z}_{ts}, \boldsymbol{z}_{t+1:T}^s \mid \boldsymbol{x}, \boldsymbol{y})$ if $z_t^{v \leftrightharpoons j_k} = \tilde{z}_{ts}$;

    **end for**

  **end for**

  Finally, we compute the ARSM gradients and update parameters:

  **for** $t = 1 : T, k = 1 : K$ *(in parallel)* **do**

    Let $\bar{f}_{tk} = \frac{1}{V} \sum_{v=1}^{V} f(z_t^{v \leftrightharpoons j_k})$;

    **for** $v = 1 : V$ *(in parallel)* **do**

      Let $g_{tk,v} = \frac{1}{K}\big(f(z_t^{v \leftrightharpoons j_k}) - \bar{f}_{tk}\big)(1 - V\pi_{tj_k})$, where $g_{tk,v}$ is the $v$th component of $\boldsymbol{g}_{tk}$;

    **end for**

  **end for**

  **for** $t = 1 : T$ *(in parallel)* **do**

    $\boldsymbol{\theta} = \boldsymbol{\theta} + \eta_{\theta}\nabla_{\boldsymbol{\theta}}\phi_t \boldsymbol{g}_t$, where $\boldsymbol{g}_t = \sum_k \boldsymbol{g}_{tk}$    with step-size $\eta_{\theta}$

  **end for**

**end while**

**Algorithm 3:** Binary-tree-ARSM policy gradient for fine-tuning a binary-tree contextual categorical sequence generation model.

---

**input** : MLE pre-trained policy parameter $\boldsymbol{\theta}$, binary code to word mapping $\nu$
**output :** Fine-tuned policy parameter $\boldsymbol{\theta}$;

---

**while** *not converged* **do**

    Receive random sample $\boldsymbol{x}, \boldsymbol{y}$;

    First, we sample a main trajectory $(z_1, \ldots, z_T)$:

    **for** $t = 1 : T$ **do**

        **for** $d = 1 : D$ **do**

            $\phi_{t,b_{t(1:d-1)}} = \mathcal{T}_{\boldsymbol{\theta}}(\boldsymbol{z}_{1:t-1}, \boldsymbol{x})_{\nu(b_{t(1:d-1)})}$

            Sample $\pi_{td} \sim \text{Uniform}(0, 1)$;

            Let $b_{td} = \mathbf{1}_{[\pi_{td} < \sigma(\phi_{t,b_{t(1:d-1)}})]}$;

        **end for**

        $z_t = \nu(b_{t(1:D)})$

    **end for**

    Second, we compute pseudo actions:

    **for** $t = 1 : T$ *(in parallel)* **do**

        **for** $d = 1 : D$ **do**

            Let $b_{td}^{(1)} = b_{td}$;

            Let $b_{td}^{(2)} = \mathbf{1}_{[\pi_{td} > \sigma(-\phi_{t,b_{t(1:d-1)}})]}$;

            **if** $b_{td}^{(1)} \neq b_{td}^{(2)}$ **then**

                If $d < D$, sample $b_{t(d+1:D)}^{(j)} \sim p_{\boldsymbol{\theta}}(b_{t(d+1:D)} \mid \boldsymbol{z}_{1:t-1}, b_{t,1:d-1}, b_{td}^{(j)}, \boldsymbol{x}), j = 1, 2$;

                Let $z_{td}^{(j)} = \nu(b_{t(1:D)}^{(j)}), j = 1, 2$;

            **end if**

        **end for**

        Let $S_t = \text{unique}(\{z_{td}^{(j)}\}_{d,j})$ which means $S_t$ is the set of all unique values in $\{z_{td}^{(j)}\}_{d,j}$. Denote the cardinality of $S_t$ as $|S_t|$, where $D \leq |S_t| \leq 2 * D$.

    **end for**

    Third, we complete sentences and evaluate the rewards for the unique set of pseudo actions:

    **for** $t = 1 : T$ *(in parallel)* **do**

        **for** $\tilde{z}_{ts} \in S_t$ *(in parallel)* **do**

            If $t < T$, sample $\boldsymbol{z}_{t+1:T}^s \sim p_{\boldsymbol{\theta}}(\boldsymbol{z}_{t+1:T} \mid \boldsymbol{z}_{1:t-1}, \tilde{z}_{ts}, \boldsymbol{x})$;

        **end for**

        **for** $d = 1 : D, j = 1 : 2$ *(in parallel)* **do**

            Let $f_{td}^{(j)} = r(\boldsymbol{z}_{1:t-1}, \tilde{z}_{ts}, \boldsymbol{z}_{t+1:T}^s \mid \boldsymbol{x}, \boldsymbol{y})$ if $z_{td}^{(j)} = \tilde{z}_{ts}$;

        **end for**

        We compute the ARSM gradients and update parameters:

        **for** $d = 1 : D$ *(in parallel)* **do**

            **if** $b_{td}^{(1)} \neq b_{tl}^{(2)}$ **then**

                Let $g_{t,b_{t(1:d-1)}} = \frac{1}{2}(f_{td}^{(1)} - f_{td}^{(2)})(1 - 2\pi_{td})$;

                $\boldsymbol{\theta}_{\text{update}} = \eta_\theta \nabla_{\boldsymbol{\theta}} \phi_{t,b_{t(1:d-1)}} g_{t,b_{t(1:d-1)}}, \quad$ with step-size $\eta_\theta$

            **end if**

            $\boldsymbol{\theta} = \boldsymbol{\theta} + \boldsymbol{\theta}_{\text{update}}$

        **end for**

    **end for**

**end while**

---

## D  Experimental Setup Details

### D.1  Image Captioning

Image captioning maps an image $x$ to a sentence $y = (y_1, \ldots, y_T)$ that summarizes the image information. It has become a standard task to compare different RL methods using policy gradient. A popular evaluation metric for this task is the CIDEr score (Vedantam et al., 2015), which measures the similarity between the generated caption $y$ and some reference ones. (Rennie et al., 2017; Anderson et al., 2018; Xu et al., 2015). We conduct our experiments on the MS COCO dataset (Lin et al., 2014) that consists of 123,287 images. Each image has at least five captions. We use the standard data split from Karpathy & Fei-Fei (2015), with 113,287 training, 5000 validation, and 5000 testing images. The vocabulary size $V$ is 9488 and the max caption length $T$ is 16. For the model architecture, we employ a Fully-Connected (FC) model without attention (Rennie et al., 2017). Image features are extracted from a pre-trained ResNet (He et al., 2016). Our implementation is based on Luo et al. (2018). We pre-train a model with MLE until convergence and use it for initialization. The CIDEr scores between the generated captions and references are used as rewards.

## E  Qualitative results

### E.1  Pseudo sentences produced by ARS-K algorithm.

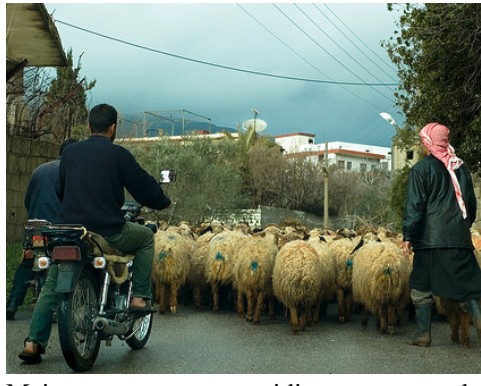

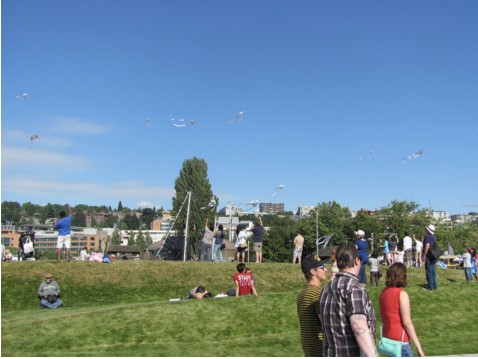

Main sentence: a man riding a motorcycle with a dog.
Pseudo sentence 1: a man riding a motorcycle with hay on it.
Pseudo sentence 2: a man riding a motorcycle with pack of sheep.

Main sentence: a group of people flying kites in a field.
Pseudo sentence 1: a group of teenagers standing in a field flying kites.

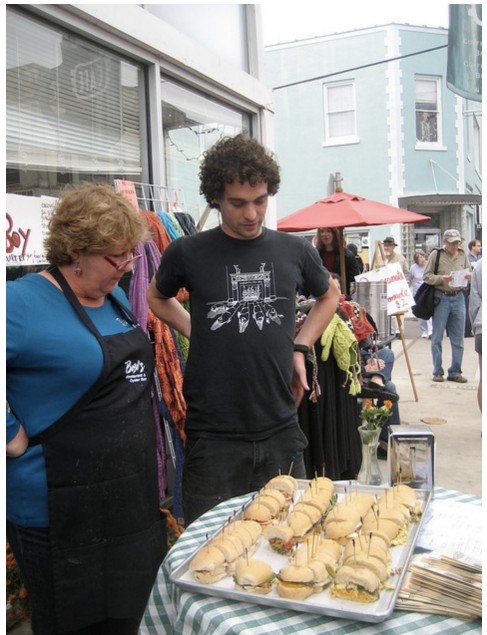

Main sentence: a man and woman are standing in a of a table.
Pseudo sentence 1: a man and woman are standing in an market.
Pseudo sentence 2: a man and woman are standing in the street.
Pseudo sentence 3: a man and woman are standing in to a tent.
Pseudo sentence 4: a man and woman are standing in front of a table.

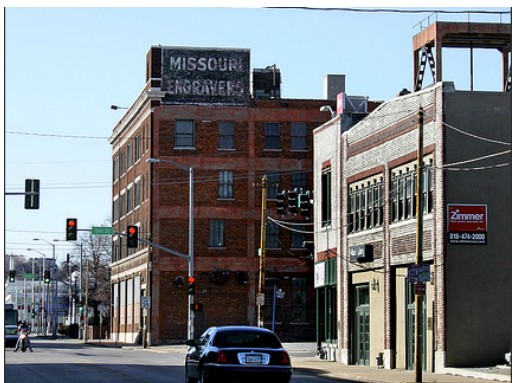

Main sentence: a city that has a large white building on it.
Pseudo sentence 1: a city with a red traffic and a large building.
Pseudo sentence 2: a city intersection with a traffic light and a street sign.
Pseudo sentence 3: a city bus is driving down the street.
Pseudo sentence 4: a city street with a city street with cars parked on it.
Pseudo sentence 5: a city road with a traffic light and a street sign.

