# OpenReview forum: "Adaptive Correlated Monte Carlo for Contextual Categorical Sequence Generation"
_ICLR.cc/2020/Conference — Accept (Poster)_

### Official Review · AnonReviewer2 · 2019-10-23
**Official Blind Review #2**

**Rating:** 8

**Review:**

The authors propose a new algorithm for unbiased stochastic gradient estimation for use in reinforcement learning of sequence generation tasks (specifically neural program synthesis and image captioning). The method consists in performing correlated Monte Carlo rollouts starting from each token in the generated sequence, and using the multiple rollouts to reduce gradient variance. An interesting property of the proposed algorithm is that the number of rollouts automatically scales with the uncertainty of the policy.

The proposed algorithm is novel, and the results are promising. Implementation of the idea seems non-trivial, but the authors provide open source code. The proposed algorithm could be impactful. The paper is clearly written.

Questions for the authors:
- Can you say anything about the optimality of scaling the number of rollouts with the policy uncertainty? Does the algorithm make optimal use of the number of rollouts? i.e. is the variance minimal for the number of rollouts, or is there scope for improvement?
- The number of rollouts being random possibly complicates efficient parallel evaluation of the rollouts (batch sizes are effectively varying). This is presumably not a problem for the chosen applications, but could you discuss the limitations in a broader setting?

**Experience Assessment:**

I have read many papers in this area.

**Review Assessment: Checking Correctness Of Derivations And Theory:**

I assessed the sensibility of the derivations and theory.

**Review Assessment: Checking Correctness Of Experiments:**

I assessed the sensibility of the experiments.

**Review Assessment: Thoroughness In Paper Reading:**

I read the paper at least twice and used my best judgement in assessing the paper.

---

> ### Author Response · Authors · 2019-11-15
> **Response to questions**
>
> Thank you for your comments and insightful questions, to which we response below. We have revised our paper accordingly and highlighted the major edits in blue.
>
> Q1: Can you say anything about the optimality of scaling the number of rollouts with the policy uncertainty? Does the algorithm make optimal use of the number of rollouts? i.e. is the variance minimal for the number of rollouts, or is there scope for improvement?
>
> Response: Since theoretical analysis of the property of the ARS-K/M estimator is very challenging and has not been carefully studied so far (except for the case of V=2, for which Yin and Zhou (2019) had proved the optimality of the ARM estimator subject to an antisymmetric constraint), in this paper we mainly focus on methodology, algorithm development, and empirical evaluation. The intuition behind the sample efficiency of using correlated Monte Carlo (MC) rollouts to reduce variance comes from antithetic sampling and variance reduction by sharing common random numbers between different expectations, as discussed in detail in “Art B. Owen. Monte Carlo Theory, Methods and Examples, chapter 8 Variance Reduction.” One of the key contributions in this paper is to study the optimal number of rollouts empirically. For example, in the neural program synthesis experiments, we compare the correlated MC rollouts with both beam search rollouts and independent MC rollouts, showing that with overall much fewer rollouts, ARSM has smaller variance and  converges to a better solution. Also, in image captioning, we show the tradeoffs between the number of rollouts (dependent on the number of reference categories K) and the gradient variance/performance. We conclude that, to get superior performance over SC, we only need as few as 10 reference categories, leading to the number of rollouts at each step around 1 on average. Up to now, we do not have any theoretical guarantees; our hypothesis is that the amount of variance reduction brought by a few correlated MC rollouts is at least on par with that by V independent MC rollouts, but we find it quite difficult to prove in theory at this moment; we suspect, there might be space for further improvement.
>
> Q2: The number of rollouts being random possibly complicates efficient parallel evaluation of the rollouts (batch sizes are effectively varying). This is presumably not a problem for the chosen applications, but could you discuss the limitations in a broader setting?
>
> Response: The number of rollouts is adaptive across different samples, different training stages, different sentence positions, and different binary tree depth. We concatenate different rollouts, and process them as a batch, whose size is varying. In our applications, since we have the access to target data, we can first pretrain our policy with MLE objective to obtain a good initial policy, so the number of unique pseudo actions is limited even when V is large (see Figure 3 (c)). Therefore, such adaptive characteristic would not affect the batch parallelization much. However, we note that in a cold-start setting where we start from a complete random policy, it is still challenging to make our methods sufficiently fast as the number of pseudo actions may be too large if V is large. We leave it as a future work to adapt our methods to this more challenging setting, where to our best knowledge, little work has been done except for Ding & Soricut (2017); d’Autume et al. (2019). We have added a discussion of this limitation in the revised paper.

---

### Official Review · AnonReviewer3 · 2019-10-23
**Official Blind Review #3**

**Rating:** 6

**Review:**

The paper presents a novel reinforcement learning-based algorithm for contextual sequence generation. The algorithm builds on the previously proposed MIXER algorithm and improves it by integrating gradient estimates with lower variance (augment-REINFORCE-swap-merge). To further improve the runtime complexity of the proposed algorithm, binary tree-based hierarchical softmax is applied. The algorithm is evaluated on the Karel dataset for neural program synthesis and the MS COCO dataset for image captioning.

The presentation of the paper must be improved (my score assumes that this will have been done). It is nice to have the detailed derivations when trying to dive deeper into the problem, but it hinders understanding of the main concepts at the first reading. Therefore, I would highly recommend to move most of the formulas to the appendix and keep instead only key ideas with intuitive explanations. It would also free some space in the main paper for the experiments from the appendix.

Several questions on the technical side:
1. Is there any intuition why pseudo actions tend to be equal to the true one when learning progresses? What causes this? Might it enforce any structure (like uniform)?
2. When all pseudo actions are the same, the gradient is zero. In theory, zero gradient of a function corresponds to its extremum. Does it mean that when all pseudo actions are the same, an extremum is reached or is it just an artifact of this particular estimator? Can one prove any results of this kind?
3. I understand that the ARSM estimator should be unbiased for V = 2. Does the estimator remain unbiased when V > 2?
4. In the experiments, the variance is shown to reduce significantly which is nice. However, in theory, does the ARMS guarantee non-increasing variance or can it potentially go up in some cases? If it can, have it ever been observed in practice?
5. How does the runtime of the proposed algorithm compare to the competitors?

Experiments:
1. Bunel et al (2018) report higher generalization on the Karel dataset. Is the difference due to the removal of the optional grammar checker? Can the same experiments be performed with this checker on or are there any constraints of the ARSM-based method?
2. The submitted code for the NPS experiment is actually the one by Bunel et al with their comments. I could not find any instructions or scripts reproducing the results of this paper (and I didn’t have much time to figure that out). One thing I wanted to check in the code is how the variance was computed for the plots?

Minor:
1. p. 4, “expected award” >> “expected reward”
2. g_{ARSM} defined twice in (5) and in the beginning of p. 4
3. “Fig. 1 (left two) plots” and “Fig. 1 (right two) plots” not good



**Experience Assessment:**

I have read many papers in this area.

**Review Assessment: Checking Correctness Of Derivations And Theory:**

I carefully checked the derivations and theory.

**Review Assessment: Checking Correctness Of Experiments:**

I assessed the sensibility of the experiments.

**Review Assessment: Thoroughness In Paper Reading:**

I read the paper thoroughly.

---

> ### Author Response · Authors · 2019-11-15
> **Point-by-point response to all questions**
>
> Thank you for your detailed comments and suggestions. We have revised our paper accordingly and highlighted the major edits in blue. Below please find our point-by-point response.
>
> On the presentation of the paper: we admit that the formulation could be intense, so we only kept those equations and definitions that are essential to understand our methods. Following your recommendation, we have further moved some equations from the main paper to the appendix, and added more intuitive explanations to facilitate the understanding of the key ideas.
>
> Response to the questions on the technical side:
>
> 1. The intuition behind the adaptiveness across training process is that as the learning progresses, the policy becomes more and more confident, meaning that the entropy of the prediction distribution at each step becomes smaller and smaller and hence the prediction probabilities would possibly only peak at fewer and fewer categories. If we look at how we get our pseudo actions z^{m swap j} at the line below Equation (4), it is clear that if the entropy of \softmax(\phi_1,...,\phi_i, …\phi_V) is small, the number of unique pseudo actions would also tend to be small.
>
> 2. It is true that if all pseudo actions are the same, the estimated ARSM gradient is zero. But if one takes another random draw, there is a certain probability that not all pseudo actions are the same, in which case the estimated ARSM gradient will not be zero. We note that the probability that all pseudo actions are the same (and hence the estimated ARSM gradient is zero) tends to increase as the training progresses. When  the policy prediction probabilities become highly peak at one category for each step, which usually only happens, if it does happen, at the end of the training process, then it is very likely that the ARSM gradient estimates will be zeros.
>
> 3. The ARM estimator is designed for V=2, and ARSM estimator is designed for V>=2. The ARSM estimator is unbiased regardless of whether V=2 or V>2.
>
> 4. We note that the variance of a gradient estimator is highly related to the parameter value at which the gradient is evaluated, and hence there is no guarantee that the variance of a gradient estimator (including ARS-K/M) is non-increasing during the training process. For example, it is totally possible that the optimal solution is associated with a large gradient variance. Note some other estimators have tried to design baselines, whose parameters are optimized to minimize the sample variance of the estimated gradients; this actually leads to a min-max optimization procedure hidden in the main algorithm that may prevent converging to a local optimal solution, as there is no guarantee that the path towards a local optimal solution will be associated with lower gradient variance.
>
> As mentioned in our paper (in NPS results and analysis part), the gradient variance at a given iteration is related to both the property of the gradient estimator and the parameter value at that iteration. It is possible that in the beginning of training, due to the poor performance of policy, the rewards can be very sparse (like in NPS), so the gradient may have lower variance compared to the later stage of training.
>
> 5. The number of rollouts at each step is an indicator of runtime. In Figures 1 and 3, we compare the number of rollouts between ARS-K/M with competitors. In Figure 1(b), we notice that in the beginning ARSM has more rollouts (slower) than RL_beam, but quickly the number of rollouts decreases and has fewer rollouts (faster) than RL_beam. In Figure 3, the effective number of rollouts of SC at each step is around 1/6 while for ARS-K the effective number of rollouts goes under 1 after 4 epochs (total of ~25 epochs of fine-tuning), meaning aside from the time for computing pseudo actions, ARS-K is at most five-times slower than SC.
>
> Response to the questions on experiments:
>
> 1. Our results differ from what’s in Bunel et al (2018) due to the exclusion of grammar checker, as in this paper we focus on the context where the action space is fixed instead of adaptive to other supervision. It is unclear at this moment how to modify ARSM to an adaptive action space, which is beyond the scope of this paper and we leave it as an interesting topic for future investigation.
>
> 2. Code: we have included the arm.sh file in the updated code folder to make it easier to reproduce our results. The variance plots are based on code in nps/train.py from Lines 418 to 422.
>
> We have addressed your minor comments.

---

### Official Review · AnonReviewer1 · 2019-10-29
**Official Blind Review #1**

**Rating:** 6

**Review:**

The paper presents experimental results on the application of the gradient ARSM estimator of Yin et al. (2019) to challenging structured prediction problems (neural program synthesis and image captioning). The authors also propose two variants, ASR-K which is the ARS estimator computed on a random sample of K (among V) labels, as well as a binary tree version in which the V values are encoded as a path in a binary tree of depth O(log(V)), effectively increasing the length of sequences to be predicted but reducing the action space at each tilmestep.

The paper is self-contained and clear. The main value of the paper is to present good experimental results on challenging tasks; the ARS-K variant, although fairly straightforward, seems to be a reasonable implementation of the ARS(M) estimator.

My main criticism on the paper is that the exact nature of the contribution is not properly stated. As far as I understand, the main value of the paper is to demonstrate the effectiveness of ASR-K/M on challenging tasks. In a first read however, it seems that the authors claim an algorithmic/theoretical contribution compared to the state-of-the-art. Comparing with the paper by Yin et al. (2019), it seems to me that the technical contribution is rather incremental (the binary tree version is a variant of the hierarchical softmax, and ASR-K seems very straightforward), up to the point that the first set of experiments is actually only about vanilla ARSM.

other comments:
- what is j in Eq 4?
- RL_beam vs ASRM on neural program synthesis: the authors say that "RL_beam overfits [...] because of biased gradients", whereas "ASRM converges to a local minimum that generalizes better". I do not see why biased gradients would help fitting the data (compared to unbiased gradients). And as far as I understood, ASRM is about getting a better gradient (hence better optimization, and hence better fitting of the data), so I really do not understand this argument.

- RL_beam vs ASRM on NPS: I do not see why ASRM cannot fit the data as well as RL_beam. Is there some regularization involved?

minor:
- "expected award" (first line section 3.1)

------ after author rebuttal

The authors answered my main concerns, I raises my score to weak accept.

**Experience Assessment:**

I have read many papers in this area.

**Review Assessment: Checking Correctness Of Derivations And Theory:**

I assessed the sensibility of the derivations and theory.

**Review Assessment: Checking Correctness Of Experiments:**

I assessed the sensibility of the experiments.

**Review Assessment: Thoroughness In Paper Reading:**

I read the paper at least twice and used my best judgement in assessing the paper.

---

> ### Author Response · Authors · 2019-11-15
> **We have clarified our contribution and addressed all the other concerns**
>
> Thank you for your constructive comments and suggestions. We have revised our paper to address your criticisms, with the major edits highlighted in blue.
>
> On the contribution of this paper: we view our core contribution as addressing the contextual categorical sequence generation problems by applying and modifying the ARS-K/M estimators. The ARSM paper only did proof-of-concepts experiments on small action spaces, while in our paper, we consider space of up to V~10^4 actions, which poses several significant challenges to efficient implementations that have been successfully addressed. For example, the swapping operations required by ARS-K/M to compute unique pseudo actions could become the computation bottleneck when V is large; to address this issue, we have developed an efficient algorithm that avoids unnecessary swapping operations to significantly accelerate the computation, as explained in detail in Appendix C.
>
> To make the concept of ARS-K/M clear and intuitive in our setting, we often describe it with correlated Monte Carlo (MC) rollouts, and interpret our method as a gradient estimation method exploiting correlated MC rollouts based token-level rewards, which naturally serve as the baselines for each other.  From this point of view, the comparison between ARS-K/M based methods and MC-K, SC together with other baselines, where either independent MC rollouts or sentence-level rewards are used, becomes clearer and demonstrates the advantage of using token-level rewards and correlated MC rollouts.  Moreover, we reveal that the number of correlated MC rollouts is automatically adapted to model uncertainty across samples, iterations, sentence positions, and depths, as illustrated in Section 4.2 and Appendix A.
>
> We hope to note that our binary tree extension is not a simple variant of hierarchical softmax. While we share with Morin & Bengio (2005) the basic idea of decomposing a full-softmax to a sequence of binary-softmax, our choices of model details are very different: We use all previous tokens (with RNN) instead of only neighboring previous tokens to compute softmax logits, and our tree construction procedure is totally data-driven using agglomerative clustering on the given word embeddings; by contrast, Morin & Bengio (2005) manually constructed a non-binary tree based on WordNet and then split the tree into binary using K-means. Further, we introduce task-specific embeddings. Our ablation studies shown in Table 2 (updated in revision) demonstrate that, using task-specific embeddings learned from the full softmax model gives superior results over off-the-shelf embeddings.
>
> Moreover, while it appears natural to combine binary softmax with ARSM, figuring out the technical and implementation details is not trivial at all, mainly because that two levels of sequential structures are involved. For example, after completing each lower-level sequence (binary code), we need to map the binary code to the higher-level vocabulary and feed the token to LSTM. Then, we use the output from LSTM to guide the generation of the binary code for the next token. Such interchange between two-level information complicates the implementation.
>
> Response to other comments:
>
> - As defined right after Eq 4, $j$ in Eq 4 is the index of a reference category randomly selected from {1,...,V}.
>
> - In the last paragraph of Section 4.1,  we have elaborated our argument about why RL_beam appears to overfit while ARSM does not.
>
> - We note ARSM tends to generate fewer unique pseudo actions as the policy becomes more certain, and provides zero gradient when zero unique pseudo action is generated, which may help provide implicit regularization. More specifically, as the ARSM estimator is unbiased, zero gradients at some iterations may imply larger gradients at other iterations, and hence our intuition is that it either freezes the parameters or update them confidently with larger gradients (more likely towards the same directions of the true gradients).

---

### Decision · Program_Chairs · 2019-12-19

**Decision:**

Accept (Poster)

**Comment:**

The paper presents a novel reinforcement learning-based algorithm for contextual sequence generation. Specifically, the paper presents experimental results on the application of the gradient ARSM estimator of Yin et al. (2019) to challenging structured prediction problems (neural program synthesis and image captioning). The method consists in performing correlated Monte Carlo rollouts starting from each token in the generated sequence, and using the multiple rollouts to reduce gradient variance. Numerical experiments are presented with promising performance.

Reviewers were in agreement that this is a non-trivial extension of previous work with broad potential application. Some concerns about better framing of contributions were mostly resolved during the author rebuttal phase. Therefore, the AC recommends publication.